# Automated Segmentation of Cystic Macular Edema in OCT B-Scan Images

**Luke Greenwood, Professor Andrew Hunter, Dr Bashir Al-Diri**
The University of Lincoln
Brayford Way, Brayford Pool, Lincoln
`luketg8@gmail.com, ahunter@lincoln.ac.uk, baldiri@lincoln.ac.uk`

**Maged Habib**
Sunderland Eye Infirmary
Queen Alexandra Rd, Sunderland
`mhabib_jc@yahoo.co.uk`

## Abstract

The analysis of retinal Spectral Domain Optical Coherence Tomography (SD-OCT) images by trained medical professionals can be used to provide useful insights into many various diseases. It is the most popular method of retinal imaging due to it's non invasive nature and the useful information it provides for making an accurate diagnosis. In this paper, we present a deep learning approach for the automating the segmentation of cystic macular edema (fluid) in retinal OCT B-Scan images. Our network makes use of atrous convolutions, skip connections, weight decay and significant image augmentation to ensure the most accurate segmentation result possible without the need for any features to be manually constructed. The network is evaluated against a publicly available dataset and achieved a maximal Dice coefficient of 95.2%, thus making it the current best performer on that dataset.

## 1   Introduction

The use of deep learning for medical purposes is on the rise and is fast becoming the methodology of choice for automated medical image analysis. [1] Deep learning is an exciting field of research, as it allows for patterns and features to be recognised from data without any human input, with resulting patterns potentially being so abstract that it be insurmountably difficult for humans to manually describe and construct features for them. [2] It creates such complex representations of data through many multiple layers of abstraction that have brought many breakthroughs to a vast multitude of fields over recent years. [3]

Optical Coherence Tomography (OCT) retinal imaging is a non-invasive technology in which high resolution cross sectional images of retinal tissue are acquired, allowing for in depth assessment and identification of abnormalities. [4] This analysis requires the skill of a trained medical professional, who would examine the images and make judgments on the features that they see present. This is naturally occasionally subject to observer error, however it is also very much a subjective area and therefore often has inter-observer variability [5], potentially culminating in misdiagnosis of patients.

The current issues that face other algorithms in this domain are that many have primarily focused on hand crafting features to assist with detecting cystic macular edema (fluid) in challenging regions of high noise or distortion in the image, or even ignoring poor quality images completely. Along with this, approaches have often had to utilise many extra techniques to group regions of fluid together as convolutions typically only take a small regions around a pixel into account when classifying it,

1st Conference on Medical Imaging with Deep Learning (MIDL 2018), Amsterdam, The Netherlands.

resulting erroneous pixels being classified as fluid. In this paper, we propose a deep learning based methodology to segment the regions of fluid within the retina, utilising image enhancements and augmentation to improve the image quality and enhance our necessary features. Coinciding with this, we use atrous convolutions in our network structure in order to evaluate a larger spatial domain when classifying individual pixels, reducing the need to perform extra grouping steps. The full automation of fluid segmentation would have many benefits, as not only does the ophthalmologist not need to manually segment the image, the quantitative nature of the results of the segmentations mean that they can easily be used to evaluate disease progression over numerous patient visits.

This paper will first discuss the work that has already been done in this field, before presenting our own proposed methodology and the results thereof. We conclude with the discussion of what we believe the future of OCT imaging to be and what we envisage the role of deep learning to be in this.

## 1.1 Literature Review

Roy et al [6] created a network, 'ReLayNet', with the purpose of both segmenting retinal layers and the fluid from OCT images. The goal of this architecture was that given a retinal OCT image $I$, each pixel will be assigned a label $l$ in the label space $L = l = \{1, ..., K\}$ for $K$ classes, such that the labels will produce a map of the OCT scan.

The structure of the network used in the paper consisted of a combination of encoder and decoder blocks are used, with the encoder blocks consisting of convolution, ReLU activation and max pooling layers along with batch normalisation, whilst the decoder blocks utilise unpooling, concatenation and convolution layers with batch normalisation and ReLU activation. Their network aimed to jointly optimise two loss functions; the multi-class logistic loss and a differentiable approximation of dice loss.

They accounted for the limited resolution and the amount of noise in the OCT images, by weighting the pixels during training. This was done such that those in close proximity to tissue-transitional regions have their gradient contributions boosted with a factor of $\omega 1$ as they are often challenging to segment. This was coupled with the boosting of pixels belonging to the retinal layers and fluid masses with a factor of $\omega 2$, due to the fact that these pixels are heavily outnumbered by background pixels. This resulting weighting scheme is finally produced;

$$\omega(x) = 1 + \omega_1 I(|\nabla l(x)| > 0) + \omega_2 I(l(x) = L) \tag{1}$$

where $I(logic)$ is an indicator function which is equal to one if $(logic)$ is true, else zero and '$\nabla$' represents the gradient operator.

The dataset used to evaluate the performance of the network was provided by Chiu et al [7], it consists of 110 annotated SD-OCT B-scan images from 10 patients with DME. However, it is important to note that this dataset only had a 58% interrater dice coefficient, indicating a reasonable amount of disagreement between the two labellers. The network was ran until convergence and the performance of the network for fluid segmentation was evaluated against the Dice overlap score, yielding a dice coefficient of 0.77. However, when the network was split into non-overlapping subsets of 8 patients for the training and 2 patients for testing and the data was trained using 8-folded cross-validation. The resulting ensemble of folded models achieved a significant improvement in results for fluid segmentation, resulting in a 0.81 Dice coefficient, thus demonstrating the potential performance benefits of combining independently trained models to produce the final network.

Lu et al [8] similarly used a deep neural network to segment the retinal fluid. Their approach used both layer segmentation using 3D graph-cut algorithms and a fully convolutional neural network to segment the regions of fluid. The results of these techniques provide differentiation between pigment epithelial detachment (PED), intra retinal fluid (IRF) and sub retinal fluid (SRF). This paper made use of relative distance maps for analysing a pixel, utilising it's distance from each segmented layer as a feature for determining the classification of fluid in the pixel. This was implemented such that for each pixel $(x, y)$ in the relative distance map, it's intensity is defined as;

$$I(x, y) = \frac{y - Y_1(c)}{Y_1(x) - Y_2(x)} \tag{2}$$

where $Y_q(x)$ and $Y_2(x)$ represent the $y$-coordinate of the 2 retinal layers.

The network itself was a modified version of the UNet [9] architecture, combined with the use of Random Forest classifiers [10] to rule out potential false positive regions, such that candidate regions

could be defined to help validate detections as being positive or negative. This was implemented with the initial assumption that pixels that could potentially be that of fluid could be found in regions of a minimum of 8-connectivity and reciprocally, pixels in regions of less than 3-connectivity could be discounted. Each region was then analysed for 16 pre defined features that could make it a candidate for a fluid region, with a label defined by $r = \frac{area(S_1 \cap S_2)}{min(area(S_1), area(S_2))}$, with $S_1$ and $S_2$ being the segmented and manually labelled regions, respectively. The candidate region was labelled as true, when $r > 0.7$. A random forest classifier for each individual fluid type was trained and the presence of fluid in volume $k$ was calculated.

Similarly, Schlegl et al [11] used a deep neural network architecture to perform automated pixel wise segmentation of IRC and SRF, with a network trained and evaluated using a large, private dataset consisting of 1200 anonymised OCT volume scans of eyes with various diseases (DME, neovascular AMD and RVO).

They decided that the distinction between the individual retinal layers was not something that was needed in this case, due to the fact that they often become obscured and difficult to segment when macular fluid is present in the scan. Therefore, the ILM and RPE layers were the only layers that were chosen to be analysed, as it had already been proven by Garvin et al [12] that these could be robustly segmented. In order to gain predictions during testing, overlapping regions of the image were extracted and a majority voting method was used such that a dense segmentation could be achieved through the labelling of a class for each pixel. The results demonstrated show that the network performed well in terms of precision, with a mean of 0.91 but struggled slightly in terms of recall, with a mean value of 0.84.

Finally, Lee et al [13] made use of pixel-wise deep learning image segmentation with an architecture similar to SegNet [14] to detect fluid in OCT images. In this case, their goal was to segment IRF to a similar standard of professional clinicians. For this detection process, macular OCT scans were extracted using an automated extraction tool from the Heidelberg Spectralis imaging database at the University of Washington Ophthalmology Department. All scans were obtained using a 61 line raster macula scan, and every image of each macular OCT was extracted. The images were labelled using a custom tool that recorded paths drawn by professionals and produced segmentations based on the boundaries drawn.

The network itself was again a modified version of UNet [9] with 18 convolutional layers and a sigmoid activation function to generate the probability map in the final layer. The model was trained using the Adaptive Moment Estimation (Adam) optimiser [15] which is a fast method of stochastic gradient descent that adapts based on lower order movements.

After training the network using 934 of the 1,289 OCT images, the process was stopped after 200,000 iterations and the model was evaluated. This yielded a maximal cross validation (on a cross validation set of 334 images) Dice coefficient of 0.911. To further validate these results the mean Dice coefficients were calculated for results against each professionally labelled dataset, showing the standard deviation between the two. This resulted in showing that the difference between the human inter-rater reliability and deep learning being 0.750 and 0.729 respectively, showing strong agreement between them. This was further enforced by there being no statistically significant difference being found between the network and the clinicians ($p = 0.247$).

Overall the approaches that have been taken so far show promise for applying deep learning to this domain, however there are ways in which we believe that progress can be made. Image processing techniques specifically targeted at improving the quality of the image and to enhance the relevant features would potentially remove the need for manually crafting features to help the network learn to detect the fluid. To complement this, non spatial related data augmentation techniques could help the network to generalise to different images. Furthermore, atrous convolutions [16] could be used to improve the fluid segmentation, as their dilated nature allows for better judgment to be made in relation to the local spatial information around the pixel. This would remove the need for manually defining features for classifying regions of the image, as the network itself would be trained to group the fluid together.

## 2 Methodology

The network we propose is an adaptation of the existing segmentation portion of the MultiNet structure from Teichmann et al [17] utilising the deep learning framework, 'TensorFlow' [18], which was in turn based upon the architecture of the FCN network [19].

### 2.1 Datasets

In order to evaluate our network accurately, we made use of a publicly available dataset to train our network, with the images in this dataset being captured using a Heidelberg Spectralis imaging device. This was acquired from Rashno et al [20], who made use of a non deep learning approach for segmenting the retinal fluid, but provided baseline results with which we can compare our algorithm. This dataset consists of 25 images from 23 patients, with scans averaging 12-19 frames with a resolution of $5.88\mu$m/pixel along the length and $3.87\mu$m/pixel along the width. The fluid regions that we took as ground truth for training and evaluating our network were segmented by 2 opthalmologists from the University of Minnesota, with an inter-rater dice coefficient score of 92.9%. In order to divide the dataset for training and validation, instead of splitting the dataset in terms of patients, we chose to take 19 images from each patient for training and 6 for validation, to ensure that the network did not overfit for any individual patient. Due to the lack of available data for training making it challenging to train the network from scratch, we opted to make use of transfer learning [21] coinciding with both image enhancement and augmentation for our training process.

### 2.2 Data Augmentation and Preprocessing

A hypothesis that we are proposing here is that the variation in image brightness, resolution and contrast in the OCT images causes problems with existing methollogies, as the effects that the resulting quality impact has on differentiating between retinal layers and fluid boundaries is significant. We therefore propose that being able to create clearer visual differentiation between retinal layers and fluid boundaries means the network will be able to produce more accurate results. We decided to test this theory, as we trained the network both with and without the use of image enhancement and data augmentation techniques to determine whether or not it provides a substantial improvement.

One approach we took to improve the suitability of the image for automated segmentation, was through the use of 'Contrast Limited Adaptive Histogram Equalisation' (CLAHE) which has been proven to improve image contrast [22]. This technique differs from typical histogram equalisation, due to it not treating the problem as global to the image. Instead this method will dynamically normalise the image's histogram to a desired clip limit, thus ensuring that the images remain as consistent as possible through the prevention of over-saturation. This helped us to produce a clearer image to analyse, thus improving the network's chance of segmenting the fluid regions accurately.

Image noise plays a big role in impacting the performance of the network's segmentation results, as it can obscure image features and can therefore make segmentation more challenging to achieve. In order to remove the noise from the images efficiently, we applied multiple preprocessing steps. One of which is a Gaussian filter [23]. This removed small fragments and noise in the image, facilitating the extraction of more significant features ie the fluid regions.

Further denoising is applied in the form of Non Local Means (NLM) smoothing [24]. This is based upon the theory that an image can be denoised through the use of a filter which observes pixels similar in colour to that of any given pixel. As this method does not simply just observe pixels in a given filter window's domain, it allows for smoothing to be based on a broader representation of the image. This method of non local filtering can be expressed as follows;

$$(p) = \frac{1}{C(p)} \int f(d(B(p), B(q))u(q)dq \tag{3}$$

where $d(B(p), B(q))$ is an Euclidean distance between image patches centered respectively at $p$ and $q$, $f$ is a decreasing function and $C(p)$ is the normalising factor.

To add an extra degree of robustness to this NLM smoothing approach, we first estimated the amount of noise in the image through averaging the results of a wavelet based estimation of the standard deviation of the Gaussian noise in the image [25]. This noise estimate is then used as the filter strength normalising factor when denoising, in an effort to preserve image details, as strong denoising filters

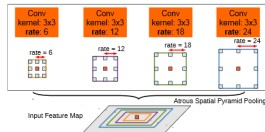

Fig. 1: Atrous Convolution for classifying the orange pixel. [16]

can remove key details from an image due to excess blurring. The data is augmented via the random applications of various image processing methods during training, consisting of random horizontal flipping, brightness and contrast adjustment and smoothing applied with varying degrees. This augmentation process allows for the network to be trained to be robust to changes in the images that it would run inference on in the future, such as changes in the image quality and brightness. [26]

## 2.3  Network Structure

The structure of the existing network that we adapted was based on the FCN architecture [19], wherein the fully connected layers of the VGG architecture [27] are transformed using an encoder and are then upscaled using a decoder to convert the encoded features into class labels.

This network originally consisted of 13 convolutional layers with pooling for the encoding blocks, these were fully convolutional and were used to encode all of the relevant, learned features that are needed to perform the segmentation into a single tensor. This encoder was coupled with 3 upsampling transposed convolution layers [28] for the decoder, that was trained to convert the features from the encoder to class labels, coinciding with skip layers being used to process higher resolution features from the lower layers. [17]

The most significant structural change was the addition of atrous convolutions [16] which we decided to implement because of the tendency of regions of fluid being grouped, as described by Lu et al [8]. We utilise these atrous convolutions with moderate rates, $r$, as we believe that their broad nature would remove the need for manually crating features for segmenting the fluid groups. This is primarily due to the fact that an atrous convolution enlarges the kernel size of a $k \times k$ filter to $k_e = k + (k-1)(r-1)$ to increase the field of view without increasing the number of parameters used, allowing for a wider context to be analysed.

To achieve this, we removed the $5^{th}$ pooling layer (the final pooling layer in the network before upsampling) from the original network to prevent loss of feature resolution at the final layer. We replaced the convolution layers in between the $4^{th}$ and $5^{th}$ pooling layers with atrous convolutions with a rate of 2. This approach was chosen based upon ideas presented by Yamashita et al [29], as they discovered that using multiple atrous convolution layers in a block arranged between the encoder and the decoder facilitates the perception of a wider context when analysing a region. In order for our network to learn to detect the fluid as a localised problem and to remove any bias based on global spatial information, we decided to remove the final two fully connected layers from the network structure. This is because we decided that the need to consider features in terms of their global positioning within the image was not needed and it could also potentially help the network to adapt to future images.

We implemented a post processing step in order to remove any potential anomalies from the results. Firstly, we applied a threshold to the resulting probability map of the network in order to discount low scoring regions and to remove any possible detections around the image border. Secondly, we applied morphological closing in order to group together any smaller regions of fluid that may have been slightly disconnected in terms of probability or by the previous thresholding process.

## 2.4  Training the Network

We trained our network using the Adam optimiser [15] with a batch size of 2, combined with a learning rate set at $1e-4$ and a weight decay [30] value set at $5e-4$. The image patches that were fed to the network during training were taken as random crops of up to $256\times256$ in size that were derived from the original image, to further remove any dependencies on the fluid's global spatial information, thus encouraging it to learn the underlying features of the fluid regions that can help to form a better understanding of what is being detected. The labels were encoded using one-hot

encoding and the loss that was used to evaluate the status of the network during training was cross-entropy;

$$loss_{class}(p,q) := -\frac{1}{|I|}\sum_{i \in I}\sum_{c \in C} q_i(c)logp_i(c) \tag{4}$$

where $p$ is the prediction, $q$ is the ground truth and $C$ is the classes. [17] Finally, to coincide with the previously discussed augmentation techniques, dropout [31] was also implemented with a probability of $0.5$ during training, ensuring that it does not overfit the data and can therefore generalise well to future images.

## 3   Results

In order to test the effectiveness of data enhancement and augmentation for improving the results in this domain, the network was trained on each of the datasets both with and without the use of these techniques. We did this through by first running the training process on the dataset provided by Rashno et al [20] (Dataset 1) for 12,000 iterations in each case.

The results of training without any augmentation being applied to the images on the validation set of Dataset 1 were positive, as shown in Table 1. It is shown here that the network was clearly able to learn the underlying pattern necessary for fluid detection and therefore was able to achieve a maximal Dice coefficient of 92.5%, with an average precision of 90.3% overall, which indicates a very high correlation between our predicted labels and that of the ground truth.

Tab. 1: Results of Training Without Denoising and Image Enhancement Techniques (Validation Set)

| - | Avg Precision | Avg Dice | Max Dice |
|---|---|---|---|
| Dataset 1 | 90.3 | 90.8 | 92.5 |

As shown in Figure 2 the network showed that it was capable of segmenting the regions of fluid well without any data augmentation being applied, performing particularly well in grouping the pixels of fluid accurately and consistently.

Unfortunately however, due to some small regions of the image appearing similar in texture to that of the fluid, the network was prone to making segmentation errors through predicting regions outside of this area as fluid, as shown in Figure 3. This is an example of a problem that we aimed to address when deciding to implement the image enhancement process into the second network. This is because this enhancement will aim to regularise the image and therefore remove the inconsistencies as best as possible, giving the network an advantage of having more prominent and distinct features in the images themselves, resulting in it having better data to learn from.

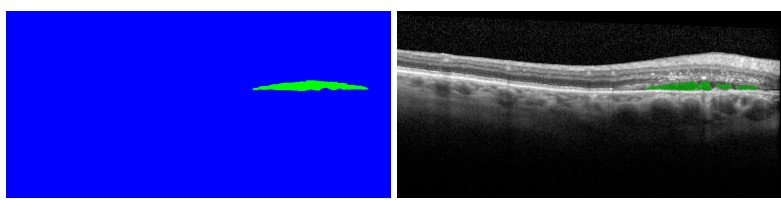

Fig. 2: Ground Truth labels (left) and results of the network with no augmentation (right)

Tab. 2: Results of Training Using Denoising and Image Enhancement Techniques (Validation Set)

| - | Avg Precision | Avg Dice | Max Dice |
|---|---|---|---|
| Dataset 1 | 90.9 | 93.6 | 95.2 |

The network that was trained using image enhancement and data augmentation, as anticipated, yielded significantly better results overall. These results are shown in Table 2, which show that we managed to achieve a maximal Dice coefficient of 95.2% and an average precision of 90.9% on

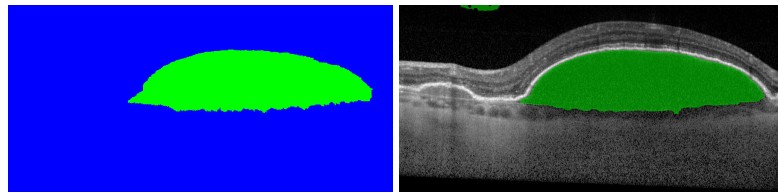

Fig. 3: The network tended to predict fluid pixels in erroneous areas of the image when no enhancement/augmentation was applied.

our validation data for Dataset 1, showing that it was able to detect a very large majority of the fluid in each B-Scan with a high degree of accuracy. This is reinforced by the results shown in Figure 4 as this shows that our network was able to segment the regions of fluid more accurately and make the distinction between the fluid like textures in the image, which is something that our other implementation without image enhancement particularly struggled with. The reasoning for this is potentially due to the fact that the enhancements on the image improved the visible contrast between these areas and the network therefore has better data to learn from to be able to determine what to predict for each pixel's label.

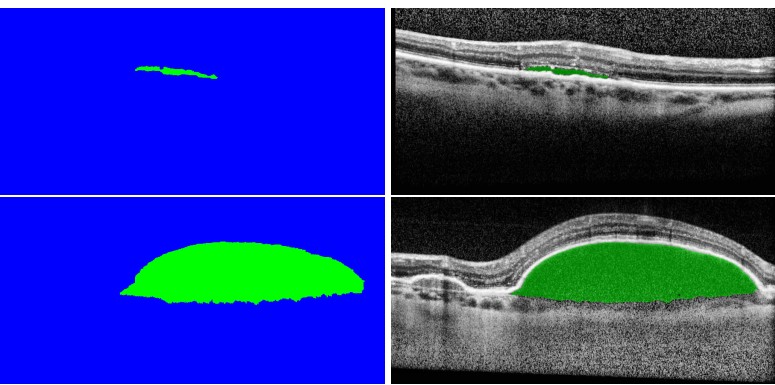

Fig. 4: Ground Truth labels (left) and results of the network with augmentation and enhancement applied (right)

However, due to some occasional artifacts and anomalies that existed within the images, the network occasionally produced false positives, thus lowering the precision of the network overall. An example of a false positive achieved can be seen in Figure 5 demonstrating the infrequent tendency of the network to over segment certain regions of the image.

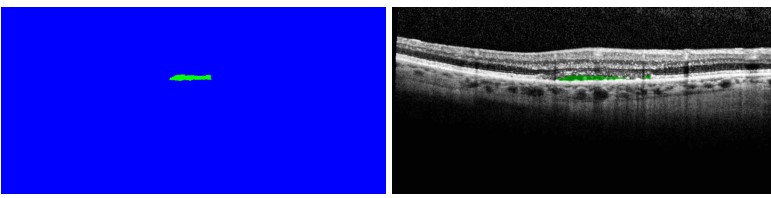

Fig. 5: The network over-segmented the image here, showing activations for a region with no fluid present

In order to make a direct comparison to the current leading pixel segmentation results on Dataset 1 achieved by the non deep learning approach of Rashno et al [20], we ran our algorithm over the entire dataset (Table 3). This analysis resulted in an average Dice coefficient of 90.4%, meaning that our implementation resulted in an improvement of 8.8% upon the current baseline, further demonstrating it's ability to segment the fluid to the highest standard.

Tab. 3: Results of Training Using Denoising and Image Enhancement Techniques (Full Set)

| - | Avg Precision | Avg Dice | Max Dice |
|---|---|---|---|
| Dataset 1 | 90.4 | 90.1 | 92.7 |

## 4   Discussion

In this paper, we have proposed a deep learning approach to fully segment the regions of fluid in 2D OCT B-Scan images, through the use of a semantic segmentation network with atrous convolutions and significant pre and post processing on the image. This method not only achieved, to the best of your knowledge, top performing results using a small amount of training data, but also converged in 12,000 iterations, demonstrating it's ability to optimise quickly to the training set. This could have potentially been far fewer iterations, had the hardware we used facilitated the use of relatively larger batch sizes during training, as the weight updates of each iteration would have been based on more examples [32] and due to our augmented and enhanced images being consistent, this could lead to a smoother convergence process. Irrespective of this, our method proposed in this paper puts forward a baseline, to which future approaches can be compared when using the dataset provided by Rashno et al [20].

We demonstrated how a large performance gain can be achieved through the use of enhancing and augmenting the images in during both training and evaluation. The reason that we decided to improve preprocessing and data augmentation stages of the original network that we were adapting, as due to the nature of OCT images being subject to noise and relatively poor resolution [33], we deemed it crucial to ensure that the image was well enhanced and denoised, such that the features we wished to segment could be given the best conditions possible. The images were largely enhanced through the use of SciPy [34], Scikit-Image [35] and OpenCV [36] libraries.

Our addition of the atrous convolutions into our network structure is also something that we attribute greatly to the success of our results, as it removed the need to manually craft features to help the network to be able to group regions of fluid together as it would take a far larger region into account when classifying a pixel. This is therefore very useful for detecting objects that are grouped and often unobscured, such as fluid.

The manual labelling of retinal fluid is something that is somewhat subjective and is therefore subject to differences of opinion between opthalmologists, as demonstrated by the low Dice coefficient between the two labellers of the Chiu et al [7] dataset. This further reinforces the need for a robust, repeatable and most importantly reliable solution to this problem to be implemented and used in clinics across the world as soon as possible. Therefore, we see the use of deep learning as a tool to be used for automated image analysis in the field of OCT imagery as a largely advantageous prospect, as combining the skill of an ophthalmologist's trained eye with a quantitative output from a deep neural network could provide an invaluable insight and can help diagnose diseases earlier and more efficiently.

In the future, the results of the individual OCT B-Scan segmentation results that we have generated can be further built upon in order to be used in a real-world clinical sense. For example, given a known scanning distance, the individual scans can be stacked and the points lying between them can be interpolated to construct a 3D model of the fluid in a given patient's scan. With this data, it is then possible for an ophthalmologist to gain an extra insight to the data in the scan, as a quantitative figure of fluid volume from each patient's visit will be produced. This could then possibly be stored beside the patient's information, thus meaning that over time, this will provide them with information on progression of diseases or effectiveness of courses of treatments.

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
