# OpenReview forum: "Automated Segmentation of Cystic Macular Edema in OCT B-Scan Images"
_MIDL.amsterdam/2018/Conference — Submitted to MIDL 2018_

### Review · AnonReviewer1 · 2018-05-04
**good presentation, weak evaluation, results reported on train/valid splits with overlapping patients**

**Rating:** 1
**Confidence:** 2

**Review:**

This paper addresses the important problem of segmenting fluid in retinal OCT scans. The authors build a fully convolutional network, substituting the last subsampling layers for atrous convolutions. The network is trained with aggressive data augmentation on a publicly available dataset, and the predictions are post-processed to improve the segmentation performance.

pros:
+ the paper is well written and easy to follow
+ concurrent models are explained with significant details

cons
- experimental validation is rather weak (only comparing to non deep learning models)
- results are reported on train/valid splits that contain images from the same patients

My main concern about the paper is the experimental evaluation. It seems there is significant amount of work dealing with the fluid segmentation problem in OCT reporting results on publicly available datasets, and applying deep learning models. However, the authors of the paper choose to use a publicly available dataset with one baseline, making it hard to assess their contribution. I do not understand why the authors don't test their method on the same dataset as [6], for example.

Another concern is related to the training/validation split. The authors argue that they choose not to split the data patient-wise, which means that they train and validate on overlapping patients. This split does not allow to assess whether the model generalizes to new patients, which is what we would want ultimately. Moreover, it is not clear how the validation split is used. Is it used to early stop the training or to test the model exclusively?

Authors mention doing transfer learning, but it is not clear how this is performed. Is the network that you use pre-trained? If so, it would be important to mention this and provide additional details.

Since cross-validation is performed, it would be beneficial to include mean and standard deviation of the results, not only average and max. Are the improvements significant?

It is not clear what running the algorithm on the entire dataset means. It seems that results were already provided on the whole dataset (with different models, doing cross-validation) in the first place.

Regarding the experiments to assess the impact of cross validation: it is well known that data augmentation acts as regularizer and might improve generalization. It would be interesting to compare the proposed model to state-of-the-art deep learning approaches.

The proposed model is very reminiscent of computer vision models such as DeepLab (and all its variants), as well as dilated residual networks. Related work substituting downsampling operations for dilated convolutions should be properly reviewed and referenced.

In the discussion, authors claim that "addition of the atrous convolution into the network is also something that we attribute greatly to the success of our results". It would be good to test this claim by training an equivalent network without dilations.

Other comments:

Numerical results of related work models should appear in the results section, with proper comparisons to the proposed model.

Figure 1 is too small.

Tables reporting results could be merged into one single table, for the sake of comparison. Please add previous state-of-the-art results to the table as well.

Figures showing results for different models (without data augmentation, with data augmentation, and ideally data augmentation without post-processing, data augmentation with post-processing) could be merged into one single figure, for the sake of easy comparison.

**Special Issue:**

No

---

### Review · AnonReviewer3 · 2018-05-08
**Easy to follow, but results and validation aren't very robust**

**Rating:** 1
**Confidence:** 2

**Review:**

This work proposed the use of a fully-convolutional network with atrous layers to perform an automatic segmentation of fluid in retinal OCT scans. The model was trained and tested on a publicly-available dataset,  and noise- and contrast-based data augmentation was found to improve results. This work combined atrous layers with the MultiNet architecture to yield good results, but the validation requires some additional experiments for this to be accepted.

Pros:
- easy to follow
- thorough literature review

Cons:
- The training/testing/validation is concerning. The training and testing set were set up to both contain images from all subjects. This doesn't really allow for a fair comparison, as the model isn't really generalizing to new data, just an extension to the data that it's already seen. Don't we want to see if it will generalize to new patients? Maybe a k-fold cross-validation between patients would give more insightful results.
- As the model is trained and tested on similar data (same patient data), it seems unfair to compare to the Rashno et al. baseline, as they used a graph cut approach with no a priori information. This comparison would likely be more fair if compared on a patient basis.
- Datasets (Section 2.1) - what transfer learning was done? It's not mentioned anywhere else.
- No standard deviation reported in results - makes it difficult to really compare performances
- Discussion makes claim that the atrous convolutions were a big contributor to the good performance - but no experiments were conducted to support this. Maybe performing an experiment without the atrous layers would be beneficial.

Some other notes:
- in-text references should appear before the period at the end of the sentence
- details about the model should be included in the abstract (i.e. FCN based on MultiNet structure, etc.)
- In "Data Augmentation and Pre-Processing" -- a figure with examples of variability might make your argument stronger
- Table 3: Why not include the baseline results for the reader?
- All tables in Results should be combined for readability
- Promising work, just needs more robust validation

**Special Issue:**

No

---

### Review · AnonReviewer2 · 2018-05-09
**Descriptions of data and methods are weak and experiments/results are not particularly novel.**

**Rating:** 1
**Confidence:** 2

**Review:**

This paper addresses the topic of segmenting fluid in retinal OCT scans, however the descriptions of experiments and data are weak and there is little novelty in what is described as well as a poor distinction between training/validation/test sets.  I cannot recommend this paper for acceptance at this time.  More detailed comments are below:

The authors provide a very lengthy description of several methods from the literature.  This is too detailed - literature should be summarised in one or two sentences per work covering the method, the data and the results.  I would also expect to see elements of how previous literature needs to be improved upon.  This is not clear to me from reading the literature review.  The author mentions that their techniqe will obviate the need for manually hand-crafted features, but I do not see that the previous works have been using such features - many of them are using deep learning.

The authors claim that Tensorflow is based upon the architecture of the FCN network, which is either worded incorrectly or a serious lack of understanding on their part.

The training and test sets are split such that data from an individual patient is contained in both.  This should never be the case in any machine learning method.  Furthermore, the data should be split into 3 sets (training, validation, test) so that network architecture and parameters can be tuned on the validation set before application to the unseen test data.  The authors here use only training and validation sets. There is no explanation of how architecture and hyper-parameters are chosen which leads to the expectation that they were tuned for best results.

The authors use data enhancement and augmentation, which they refer to as transfer learning, although transfer learning in fact refers to using networks pretrained on different datasets.

The description of the network structure (section 2.3) is difficult to understand, particularly the first 2 paragraphs.  Details of the thresholding (how was the threshold chosen?) and morphological closing (what size kernel?) should be provided.
The size of the original images in the X-Y dimensions is not provided.

The results section refers to "each of the datasets" although only one dataset has been described thus far.  The meaning is unclear. It further refers to "dataset 1" and "the entire dataset", which is very confusing and I cannot determine what data has been used or whether there is crossover between training and test data at any point.

The authors compare results with and without data enhancement/augmentation and determine that these techniques improve the performance, which is not entirely unexpected, but the differences are extremely small and there can be no suggestion that they are significant.  This would not be an exceptionally novel finding in any case.
I cannot comment on the comparison with the results of Rashno et al. since it is unclear to me whether this involves crossover of training and test data (it refers to "Full Set" in the table as opposed to "Validation set" in other tables).  Furthermore, Rashno et al did not use a deep learning method, and it would not be particularly novel for a deep learning method to outperform a conventional method.  There are many deep learning methods on this type of data that have been published, and it would be more appropriate to compare results with those.

**Special Issue:**

No

---

### Decision · Program_Chairs · 2018-05-15
**Paper57 Acceptance Decision**

Reject